# Assessment and Training of Perceptual-Motor Function: Performance of College Wrestlers Associated with History of Concussion

**DOI:** 10.3390/brainsci14010068

**Published:** 2024-01-10

**Authors:** Gary B. Wilkerson, Lexi R. Fleming, Victoria P. Adams, Richard J. Petty, Lynette M. Carlson, Jennifer A. Hogg, Shellie N. Acocello

**Affiliations:** 1Department of Health & Human Performance, University of Tennessee at Chattanooga, Chattanooga, TN 37403, USA; lynette-carlson@utc.edu (L.M.C.); jennifer-hogg@utc.edu (J.A.H.); shellie-acocello@utc.edu (S.N.A.); 2Department of Intercollegiate Athletics, Lincoln Memorial University, Harrogate, TN 37752, USA; lexirfleming@gmail.com; 3Sports Medicine Outreach Program, Piedmont Physicians Athens Regional Medical Center, Watkinsville, GA 30677, USA; victoria.adams@piedmont.org; 4Department of Intercollegiate Athletics, University of Tennessee at Chattanooga, Chattanooga, TN 37403, USA; rpetty@tennessee.edu

**Keywords:** reaction time, neural efficiency, intra-individual variability, performance enhancement, mild traumatic brain injury, virtual reality

## Abstract

Concussion may affect sport performance capabilities related to the visual perception of environmental events, rapid decision-making, and the generation of effective movement responses. Immersive virtual reality (VR) offers a means to quantify, and potentially enhance, the speed, accuracy, and consistency of responses generated by integrated neural processes. A cohort of 24 NCAA Division I male wrestlers completed VR assessments before and after a 3-week VR training program designed to improve their perceptual-motor performance. Prior to training, the intra-individual variability (IIV) among 40 successive task trials for perceptual latency (i.e., time elapsed between visual stimulus presentation and the initiation of movement response) demonstrated strong discrimination between 10 wrestlers who self-reported a history of concussion from 14 wrestlers who denied ever having sustained a concussion (Area Under Curve ≥ 0.750 for neck, arm, and step movements). Natural log transformation improved the distribution normality of the IIV values for both perceptual latency and response time (i.e., time elapsed between visual stimulus presentation and the completion of movement response). The repeated measures ANOVA results demonstrated statistically significant (*p* < 0.05) pre- and post-training differences between groups for the IIV in perceptual latency and the IIV in response time for neck, arm, and step movements. Five of the six IIV metrics demonstrated a statistically significant magnitude of change for both groups, with large effect sizes. We conclude that a VR assessment can detect impairments in perceptual-motor performance among college wrestlers with a history of concussion. Although significant post-training group differences were evident, VR training can yield significant performance improvements in both groups.

## 1. Introduction

Wrestling demands fast reactive responses to an opponent’s actions for performance success [1,2,3], and it is among the sports with the highest incidence of concussion [4]. A subtle impairment in the brain’s information processing efficiency may be a long-term effect of concussion that contributes to an increased susceptibility to a subsequent concussion [5], as well as increased risk of musculoskeletal injury [6,7]. Training activities designed to optimize the functional capabilities of athletes have traditionally focused on the improvement of physical attributes, such as muscle strength, power, endurance, and flexibility [8,9]. Few studies have assessed the potential benefit of “cognitive training” for performance enhancement or injury prevention [10,11,12]. Despite compelling evidence of the importance of brain processing efficiency for the generation of rapid, accurate, and consistent motor responses [8,12,13,14,15], the best means to assess and train cognitive aspects of sport performance remains unclear [16].

Dual-task activities that impose simultaneous cognitive and motor demands are commonly used to assess the effect of the cognitive load on the performance of a motor task, such as postural balancing or walking gait [17]. The extent to which the neural circuits activated by the two tasks are segregated versus integrated may have important differential effects on performance [18]. For example, a non-visual cognitive task that requires an internal focus of attention on abstract thought and reasoning (e.g., n-back working memory task) has been shown to suppress visuospatial attention, the processing of perceptual input, and motor responsiveness [19,20,21]. Rather than simultaneously administering distinctly different cognitive and motor tasks, a visuospatial task that simultaneously engages perceptual, cognitive, and motor processes for the achievement of a whole-body functional goal may have greater relevance to sport performance and injury resistance [12,13,15,16,22].

The combination of the visual detection and cognitive interpretation of environmental stimuli is often referred to as a “perceptual-cognitive skill” [12,15,16], whereas the term “cognitive-motor skill” has been used to refer to the interrelated neural processes involved in decision-making and the execution of complex movement responses [18,23,24]. Because cognition is integral to both perception and motor programming, the term “perceptual-motor performance” may be used to represent the observable result of integrated stimulus–response processing [13,16]. Immersive virtual reality (VR) offers a means to acquire numerous measurements during the performance of a perceptual-motor task that requires complex whole-body responses to visual stimuli [25,26].

An integrated multi-component VR assessment may identify individuals who would otherwise be exposed to an unrecognized elevation of injury risk [27], and participation in a training activity that incorporates a similar approach may yield beneficial improvements in perceptual-motor performance. Thus, the purposes of this study were: (1) to assess the potential for VR measurements of perceptual-motor performance to discriminate college wrestlers with a history of concussion from those who deny having ever sustained a concussion, and (2) to assess the potential for perceptual-motor performance improvements after six VR training sessions are completed over a 3-week period. 

## 2. Materials and Methods

We used immersive VR to administer an integrated perceptual, cognitive, and motor challenge that quantified the speed, accuracy, and consistency of simultaneous neck rotation, arm reaching, and single step lunging movements in response to successive presentations of distinct types of moving visual stimuli that required cognitive interpretation for response selection. The investigational device used in this study is not yet approved by the FDA for any purpose. Each participant confirmed or denied a history of having sustained a concussion by responding to an electronic survey question acquired prior to a VR pre-training assessment. We administered pre- and post-training assessments, as well as a 3-week VR training program, during regularly scheduled sessions that included all participants. All participants wore wrestling shoes when engaged in the assessment and the training activities were performed on a wrestling mat. We assessed the discriminatory power of the pre-training VR metrics using receiver operating characteristic analyses. We used repeated measures ANOVA to assess the group differences (2 groups), pre- to post-training changes (2 sessions), and any group by session interaction.

### 2.1. Participants

A cohort of 24 NCAA Division I male wrestlers (20.5 ± 1.8 years, 17.6 ± 0.07 m, 79.5 ± 11.8 kg) representing all weight classes agreed to participate in a 3-week program of immersive VR training during the month preceding the initiation of regular pre-season practice sessions, and each participant voluntarily provided electronic survey responses to items pertaining to injury history. The Institutional Review Board of the University of Tennessee at Chattanooga approved all the study procedures, including the method used to document informed consent. The athletes were advised that participation was voluntary rather than mandatory for all team members. The only exclusionary criterion was an injury-related limitation in the ability to perform rapid single-step lunging movements. 

### 2.2. Procedures

Prior to testing, an eye calibration procedure ensured the accurate measurement of eye position in relation to the visual stimuli presented on the VR headset display (PICO Neo3 Pro Eye, PICO Immersive, Ltd., 123 Mountain View, CA, USA). A T-pose position (i.e., standing upright with both arms abducted to 90° and elbows fully extended) was used to acquire a measurement of the distance between the hand controllers, which was used to calibrate the positions of the left and right virtual response targets at 30% beyond the maximum arm reach distance and outside the peripheral field of view (i.e., eye and neck movements required to visualize a virtual response target). The pre- and post- training assessments involved a series of 40 successive arm reaching and whole-body lunge-step movements in the left or right directions (Figure 1, Appendix A) according to the stimulus–response instructions for visual stimuli that moved horizontally across the black background of the VR headset display (Figure 2). The correct response to the appearance of a filled white circle was to execute a reaching/lunging movement in the same direction as the circle motion (i.e., a congruent response) so that the hand controller made contact with a virtual response target (i.e., a green spherical object). The correct response to the appearance of a moving white ring was to execute a reaching/lunging movement in the opposite direction to the ring motion (i.e., an incongruent response) so that the hand controller made contact with a virtual response target. A visual stimulus initially appeared in either a central location, with motion toward either a left or right peripheral position, or it initially appeared in a left or right peripheral location, with motion toward a central position. Both hand controller vibration and an auditory tone coincided with contact with the virtual response target.

The time elapsed from stimulus appearance to the initiation of a given body movement in the correct direction (i.e., 6 degrees of neck rotation, 10 cm of hand controller displacement, or 10 cm of step displacement) defined the perceptual latency (PL). The time elapsed from stimulus appearance to movement response completion (i.e., maximum neck rotation, maximum arm reach, or single-step lunge displacement) defined the response time (RT). We calculated the 40-trial average values for the PL (PL-Avg) and RT (RT-Avg) for the simultaneously performed neck, arm, and step movements (lower values represent better performance), as well as the intra-individual variability values (i.e., 40-trial standard deviation; lower values represent better performance) for the PL (PL-IIV) and RT (RT-IIV). We assessed the speed–accuracy trade-off through the calculation of a rate correct score (RCS) derived from the number of correct responses divided by the 40-trial sum of the elapsed time for arm movements (higher values represent better performance) for both the PL (PL-RCS) and RT (RT-RCS). Intra-class correlation coefficients demonstrating acceptable test–retest reliability has previously been documented for the various neck, arm, and step VR metrics, with the PL-Avg and RT-Avg values ranging from 0.837 to 0.922, the PL-IIV and RT-IIV values ranging from 0.693 to 0.836, and the PL-RCS and RT-RCS values of 0.851 and 0.887, respectively [28].

The VR perceptual-motor training program consisted of 2 sets of 20 trials during each session, which were performed 2 times per week over a period of 3 weeks. The key difference from the VR assessment procedure was the lack of a virtual response target, which eliminated the need for neck rotation and arm reaching (Figure 3). Starting from a staggered stance (i.e., a self-selected foot positioned in advance of the other foot), the wrestlers simply executed a lunging movement. The wrestlers received visual and auditory confirmation of the correct versus incorrect directional movement responses. We used an early version of the VR training software (Perceptual Response Training Version 1.0, REACT Neuro, Cambridge, MA, USA, 2022) that did not record the performance data during the training sessions. Because the training was an element of scheduled team activities, 100% compliance was attained.

### 2.3. Data Analysis

To assess the potential for discrimination between wrestlers with a history of concussion (HxC) from those without a history of concussion (NoC), we performed receiver operating characteristic (ROC) analyses for each measure of the pre-training perceptual-motor performance. Area under the curve (AUC) values were interpreted as acceptable in the range of 0.70 to 0.79, and excellent if ≥0.80 [29]. Only those metrics that exhibited an AUC ≥ 0.70 were further analyzed for the determination of the optimal cut-off point for conversion into a binary categorization of high-risk versus low-risk, which was based on Youden’s Index for maximum discrimination. We then performed cross-tabulation analyses to assess the statistical significance of exposure–outcome associations (Fisher’s Exact One-Sided *p*) and to calculate the classification accuracy statistics, including sensitivity, specificity, and an odds ratio (OR) with a 95% confidence interval (CI) for each potential predictor. The interpretation of the OR magnitude as small, medium, or large corresponded to values of 1.32, 2.38, and 4.70, respectively [30]. The intrinsic credibility of the OR was assessed by comparing it to the 95% skepticism limit, which represents both the OR magnitude uncertainty and the margin by which it excludes a null effect [31]. We assessed the normality of the data distributions using the Shapiro–Wilk test for each measure. We used natural log (Log_e_) transformation to improve the normality of each distribution that demonstrated a statistically significant (*p* < 0.05) positive skew (Appendix A). We used repeated measures analysis of variance (ANOVA) to assess the statistical significance (*p* < 0.05) of a difference between groups (HxC versus NoC), a difference between sessions (pe- versus post-training), or the existence of a group by session interaction for each VR measure. Because the study was exploratory, no correction for multiple comparisons was used [32]. We interpreted a partial eta-squared (η_p_^2^) value ≥ 0.14 as a large effect [33]. We performed all the analyses using IBM SPSS version 29.0 (IBM Corporation, Armonk, NY, USA, 2022).

## 3. Results

Among the 42% of wrestlers who self-reported HxC (10/24), a single concussion was reported by six, and ≥2 concussions were reported by four. The median time since the most recent concussion occurrence was 22 months (range: 9 to 120 months). The VR metrics that demonstrated the strongest discriminatory power (AUC ≥ 0.750) were the PL-IIV values for neck, arm, and step movements (Table 1). The ROC curves for PL-IIV were remarkably similar for the neck, arm, and step movements (Figure 4). The binary classifications derived from Youden’s Index yielded OR values that were both statistically significant and intrinsically credible for five of the six PL-IIV and RT-IIV metrics. 

Natural log transformation improved the distribution normality of the PL-IIV, RT-IIV, and PL-Avg values for the neck, arm, and step movements, and seven of the nine metrics demonstrated significant pre- to post-training improvement in both groups and with large effect sizes (Table 2). We found significant differences in the Log_e_ PL-IIV between the HxC and NoC groups, but similar patterns of pre- to post-training change were evident for the neck, arm, and step movements (Figure 5). The distribution normality was adequate without transformation for the neck, arm, and step RT-Avg, as well as the PL-RCS and RT-RCS for the arm movements. A statistically significant improvement with a large effect size was evident for four of the five metrics, but none of them demonstrated a statistically significant difference between groups (Table 3).

## 4. Discussion

Our findings demonstrate that an integrated multi-component task administered using an immersive VR system can detect impaired perceptual-motor function among college wrestlers who self-report a remote history of concussion. Furthermore, our findings support the potential for immersive VR training to improve the speed, accuracy, and consistency of perceptual-motor responses, regardless of whether a wrestler has reported a history of concussion. Response time has previously been found to be a key factor in competitive wrestling success [1,2,3], but very little research has addressed the speed–accuracy trade-off or response consistency, and almost no evidence exists to inform efforts to promote favorable neuroplastic adaptations [11].

An understanding of the neural mechanisms that underlie the effective execution of goal-directed behaviors is probably the best guide for the design of tests and training activities [34]. Information processing by the brain involves the synchronization of signals conveyed through neural circuits that create transient functional integration of spatially separated network nodes [35,36]. Rapid disengagement and reconfiguration of the neural circuits within and between brain networks creates signal variability that reflects an individual’s information processing capacity [36,37]. For example, responsiveness to an event in the external environment requires the suppression of activity in the internally focused default mode network (DMN) and the upregulation of activity in the executive control network (ECN), which is modulated by the salience network (SN) when a perceived stimulus is interpreted as relevant to a behavioral goal [38,39,40]. The increased neural signal variability created by such switching between brain states directly relates to the speed, accuracy, and consistency of behavioral responses to external stimuli [41,42,43]. Conversely, lapses in externally focused attention have been attributed to the excessive activation of nodes within the DMN [38,44], which can produce inconsistent responses to external stimuli (i.e., elevated IIV). 

Assessment and training activities that focus on fast reactive responses to relatively “simple” visual stimuli may have limited relevance to sport performance capabilities, whereas activities that impose task-relevant cognitive demands (e.g., the inhibition of prepotent responses, discrimination between distinct types of visual stimuli, and decision-making that requires movement responses) may provide better ecological validity [8,15,22]. Previous studies have demonstrated that the measures of IIV for successive choice responses (e.g., SD, coefficient of variation, mean squared successive differences, or multiscale entropy) are more sensitive than measures of central tendency (e.g., mean or median) for the detection of impaired neural function, which has been confirmed using advanced diagnostic technologies [44,45]. Furthermore, the existence of an inverse relationship between brain signal variability and behavioral performance variability has been established by previous research [37,38,41,42], which is more evident during the performance of a cognitively challenging task [42]. Thus, the PL-IIV and RT-IIV differences we documented between the HxC and NoC groups strongly support the value of performance consistency (i.e., low behavioral IIV) as an indirect indicator of the perceptual-motor processing efficiency within the brain [35,38]. Furthermore, our finding of significant post-training IIV decreases for both groups suggests that favorable neuroplasticity was induced by the immersive VR training program, which may promote the resolution of long-term concussion effects, reduce the risk of sport-related injury, and enhance sport performance capabilities.

The limitations of this study include its observational design, which did not include the random assignment of participants to a control group, and the close similarity of the training activity to the method we used to assess the performance improvements. A near-transfer effect refers to improvement in the performance of a task that was a key component of the preceding training activity, whereas a far-transfer effect refers to a post-training improvement in some objective indicator of a related sport-specific skill level or a real-world sport performance outcome [11,46]. Because arm reaching was not a component of the training activity performed by the wrestlers, the statistically significant improvements in the PL-IIV, RT-IIV, PL-Avg, RT-Avg, PL-RCS, and RT-RCS for arm reaching suggest that some neural adaptation induced by the single-step lunging responses had an upper extremity carryover effect. More stringent standards for the documentation of far-transfer, such as the points scored during a specified amount of competitive performance time or the avoidance versus occurrence of an injury during a specified surveillance period, should be considered for further research on the potential benefits of perceptual-motor training. Self-reporting of concussion history is often identified as a study limitation, but we cross-referenced the survey responses with the medical records maintained by the athletic program to confirm their accuracy. Potential confounding factors that we did not address, such as sleep disruption, depression, anxiety, and any chronic health disorders, should be included as covariates in future research.

The early version of the VR training software used for this study did not permit a progressive increase in the task difficulty over the course of the program, which may be a factor that could maximize the perceptual-motor performance gains [9,10,11]. The low volume of training (i.e., a total of only 12 sets of 20 repetitions of the VR task) was a consequence of the limited VR equipment availability and time constraints imposed by the wrestling team’s schedule. Despite these limitations, we observed substantial performance improvements of similar magnitude in both groups of college wrestlers. The volume of perceptual-motor training required to realize the maximum benefit remains unknown, as well as the frequency of training sessions required to retain performance gains [11]. In addition to concussion history, important factors to consider that may influence training effects include sex, age, sport, and level of competition. Future research should also assess the potential for training to enhance an athlete’s ability to maintain the optimal perceptual-motor performance in a fatigued state [11]. 

We consider our most important findings to be the strong associations between a remote history of concussion and both PL-IIV and RT-IIV among college wrestlers, along with the significant post-training improvements in perceptual-motor performance for both the HxC and NoC groups. Recent research has produced compelling evidence that concussion has a long-term effect that reduces brain signal variability, which adversely affects the ability to respond to unpredictable environmental demands [45]. Thus, the inverse relationship between brain signal variability and perceptual-motor performance consistency (i.e., low PL-IIV and low RT-IIV) may have strong clinical value as an indicator of brain health [45]. Because the standard clinical tests appear to be insufficiently sensitive for the detection of subtle concussion effects, and the current clinical guidelines lack specific recommendations for rehabilitation [46], many athletes may possess an unrecognized brain processing impairment that adversely affects their performance capabilities and susceptibility to injury for months or years following concussion. Although an immersive VR system is necessary for the clinical application of this study’s results, the cost of such equipment is far more affordable than the diagnostic neuroimaging methods that are currently required for the detection of subtle impairment. Our findings support the utilization of immersive VR to administer a challenge that requires the integration of perceptual, cognitive, and motor processes for both the assessment and training of competitive athletes.

## 5. Conclusions

Both wrestling performance success and the avoidance of musculoskeletal injuries are heavily dependent on the neural processing efficiency within the brain, which we believe can be estimated by the speed, accuracy, and consistency of measurable behavioral responses to events displayed within an immersive VR environment. The intra-individual variability in successive responses to visual stimuli appears to be a particularly important indicator of neural efficiency (i.e., low behavioral IIV) versus neural impairment (i.e., high behavioral IIV). Any competitive athlete may derive benefit from a training program designed to enhance their perceptual-motor performance, but such training may be particularly important for an athlete who possesses an undetected residual impairment from a prior concussion. Immersive VR assessment and training may provide a valuable addition to the traditional methods used to enhance their performance capabilities and mitigate injury risk. Ideally, all athletes on a given team would be tested to identify any immersive VR performance deficiency, which might be reduced through training that integrated perceptual, cognitive, and motor demands.

## Figures and Tables

**Figure 1 brainsci-14-00068-f001:**
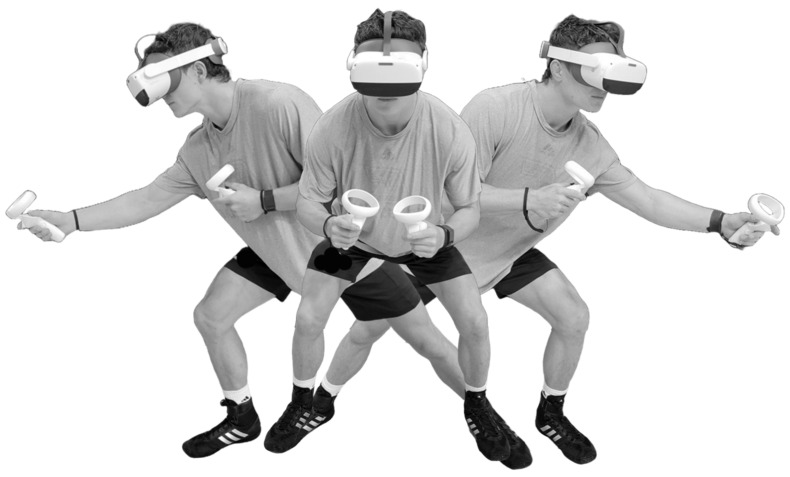
Immersive virtual reality assessment: ready position (center) prior to appearance of a moving visual stimulus, which was followed by simultaneous neck rotation, arm reaching, and single-step lunging toward virtual response target in right or left direction.

**Figure 2 brainsci-14-00068-f002:**
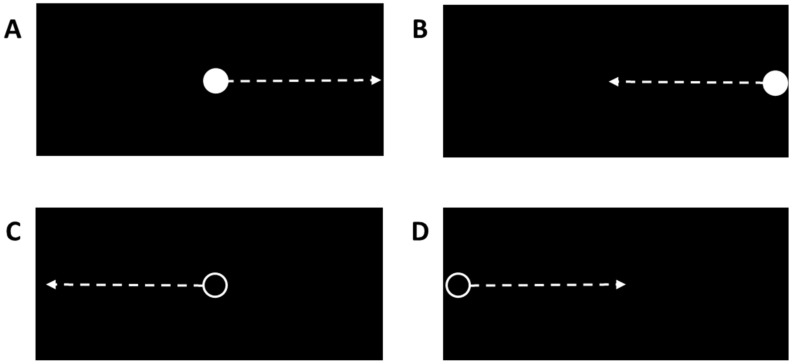
Examples of distinct types of moving visual stimuli displayed on immersive virtual reality headset display: (**A**) Congruent stimulus (filled white circle) initially appearing in a central position on display; (**B**) congruent stimulus (filled white circle) emerging from peripheral margin of display; (**C**) incongruent stimulus (white ring) initially appearing in a central position on display; (**D**) incongruent stimulus (white ring) emerging from peripheral margin of display.

**Figure 3 brainsci-14-00068-f003:**
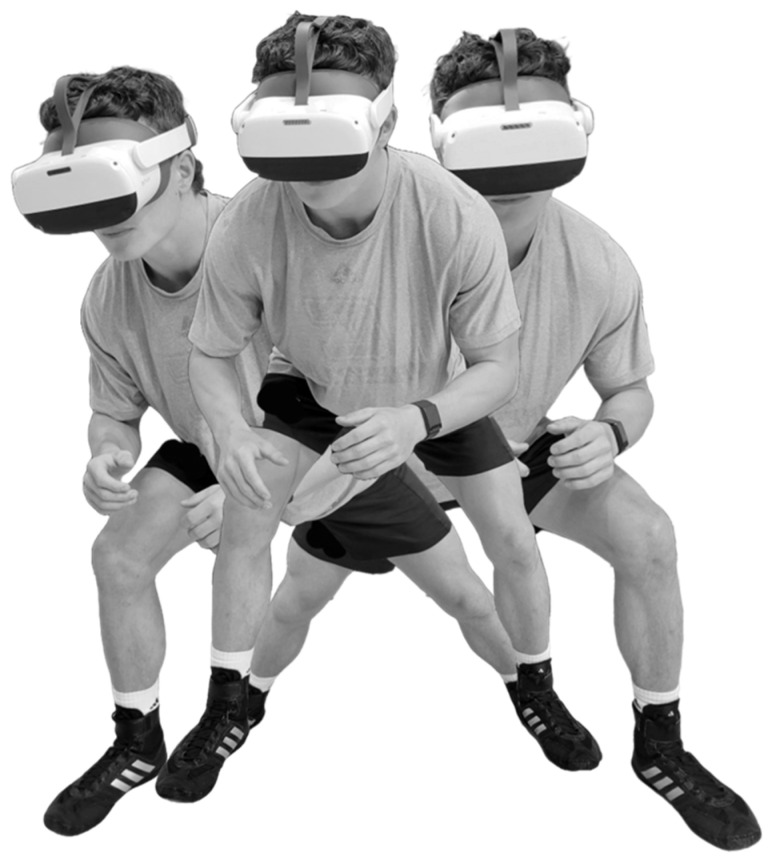
Immersive virtual reality training: ready position prior to appearance of moving visual stimulus, which was followed by a single-step lunging movement in right or left direction according to the visual stimulus type (congruent filled white circle or incongruent white ring) and its direction of motion across the headset display.

**Figure 4 brainsci-14-00068-f004:**
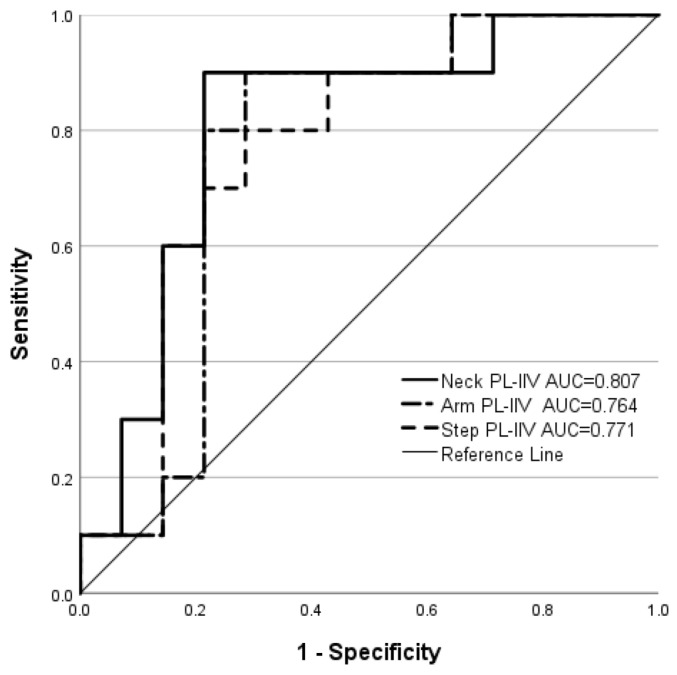
Receiver operating characteristic curves for discrimination of history of concussion from no concussion history based on neck, arm, and step perceptual latency intra-individual variability (PL-IIV).

**Figure 5 brainsci-14-00068-f005:**
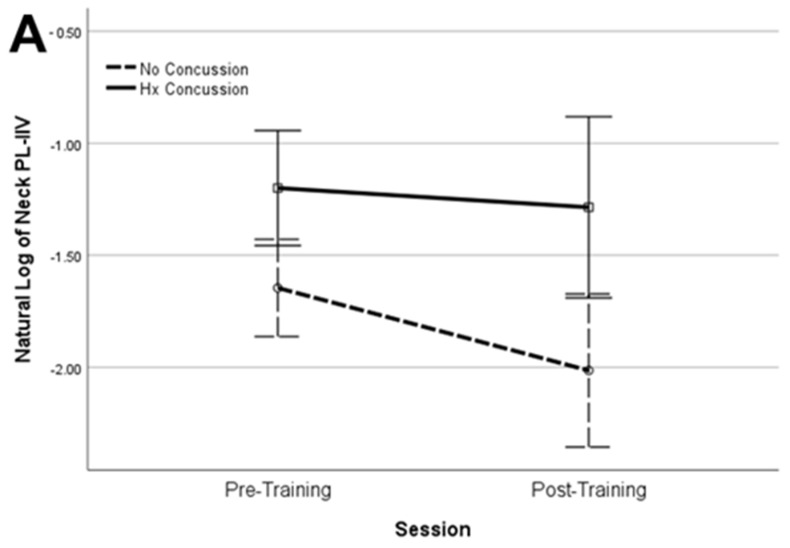
Pre- to post-training change in neck (**A**), arm (**B**), and step (**C**) perceptual latency intra-individual variability for history of concussion (solid line) and no concussion (dashed line) groups (error bars represent 2 × standard error).

**Table 1 brainsci-14-00068-t001:** Discriminatory power for history of concussion (HxC) versus no concussion (NoC) based on intra-individual variability in perceptual latency (PL-IIV) and response time (RT-IIV).

Metric	AUC	Cut-Off Point	*p* *	Sensitivity	Specificity	OR	95% CI	SL
Neck PL-IIV	0.807	≥0.215	0.001	0.90	0.79	33.00	2.91, 374.31	10.43
Arm PL-IIV	0.764	≥0.217	0.004	0.90	0.71	22.50	2.11, 240.48	15.98
Step PL-IIV	0.771	≥0.215	0.018	0.80	0.71	10.00	1.44, 69.26	20.43
Neck RT-IIV	0.793	≥0.241	0.001	0.90	0.79	33.00	2.91, 374.36	10.43
Arm RT-IIV	0.771	≥0.247	0.007	0.80	0.79	14.67	1.97, 109.20	9.58
Step RT-IIV	0.775	≥0.242	0.007	0.80	0.79	14.67	1.97, 109.20	9.58

* Fischer’s exact one-sided test. AUC: Area under curve. OR: Odds ratio. 95% CI: 95% confidence interval. SL: Skepticism limit.

**Table 2 brainsci-14-00068-t002:** Pre- to post-training change in intra-individual variability in perceptual latency (PL-IIV) and response time (RT-IIV), and 40-trial perceptual latency average (PL-Avg) among athletes reporting no concussion (NoC) versus history of concussion (HxC).

Metric	Group	Geometric Mean (Log_e_ Mean)		Group × Session Interaction		Group Difference		Session Difference	
		Pre-Training	Post-Training	*p*	η_p_^2^	*p*	η_p_^2^	*p*	η_p_^2^
Neck PL-IIV *	NoC	0.193 (−1.646)	0.133 (−2.015)	0.231	0.065	0.005	0.303	0.060	0.152
	HxC	0.301 (−1.200)	0.276 (−1.286)						
Arm PL-IIV *	NoC	0.199 (−1.615)	0.133 (−2.018)	0.339	0.042	0.005	0.301	<0.001	0.414
	HxC	0.294 (−1.225)	0.231 (−1.468)						
Step PL-IIV *	NoC	0.207 (−1.576)	0.153 (−1.876)	0.668	0.009	0.008	0.282	0.005	0.303
	HxC	0.317 (−1.148)	0.253 (−1.374)						
Neck RT-IIV *	NoC	0.222 (−1.505)	0.154 (−1.872)	0.221	0.067	0.007	0.285	0.022	0.216
	HxC	0.296 (−1.216)	0.263 (−1.335)						
Arm RT-IIV *	NoC	0.214 (−1.543)	0.136 (−1.995)	0.073	0.139	0.003	0.336	<0.001	0.410
	HxC	0.285 (−1.256)	0.243 (−1.415)						
Step RT-IIV *	NoC	0.220 (−1.513)	0.155 (−1.863)	0.272	0.055	0.009	0.269	<0.001	0.448
	HxC	0.302 (−1.199)	0.246 (−1.401)						
Neck PL-Avg *	NoC	0.609 (−0.495)	0.567 (−0.567)	0.203	0.073	0.078	0.134	0.714	0.006
	HxC	0.632 (−0.459)	0.657 (−0.419)						
Arm PL-Avg *	NoC	0.718 (−0.331)	0.603 (−0.505)	0.925	<0.001	0.301	0.049	<0.001	0.701
	HxC	0.760 (−0.275)	0.641 (−0.444)						
Step PL-Avg *	NoC	0.709 (−0.344)	0.613 (−0.489)	0.513	0.020	0.182	0.080	<0.001	0.545
	HxC	0.750 (−0.288)	0.671 (−0.400)						

* Log_e_ transformation improved distribution normality; Loge SD cannot be transformed back into original measurement units.

**Table 3 brainsci-14-00068-t003:** Pre- to post-training change in response time average (RT-Avg) and rate correct score for arm perceptual latency (PL-RCS) and arm response time (RT-RCS) among athletes reporting no concussion (NoC) versus history of concussion (HxC).

Metric *	Group	Mean ± SD		Group × Session Interaction		Group Difference		Session Difference	
		Pre-Training	Post-Training	*p*	η_p_^2^	*p*	η_p_^2^	*p*	η_p_^2^
Neck RT-Avg	NoC	0.931 ± 0.135	0.882 ± 0.116	0.075	0.137	0.354	0.039	0.725	0.006
	HxC	0.934 ± 0.110	0.968 ± 0.134						
Arm RT-Avg	NoC	1.129 ± 0.151	1.018 ± 0.154	0.606	0.012	0.240	0.062	<0.001	0.529
	HxC	1.187 ± 0.144	1.097 ± 0.126						
Step RT-Avg	NoC	1.192 ± 0.159	1.096 ± 0.132	0.131	0.101	0.219	0.068	0.043	0.173
	HxC	1.213 ± 0.112	1.198 ± 0.110						
ARM PL-RCS	NoC	1.18 ± 0.44	1.56 ± 0.33	0.351	0.040	0.412	0.031	<0.001	0.601
	HxC	1.11 ± 0.37	1.38 ± 0.42						
Arm RT-RCS	NoC	0.77 ± 0.23	0.94 ± 0.17	0.190	0.077	0.305	0.048	<0.001	0.424
	HxC	0.73 ± 0.19	0.81 ± 0.22						

* Log_e_ transformation did not improve distribution normality, analysis of original data.

## Data Availability

The data presented in this study are available from the corresponding author upon request. The data are not publicly available due to concern for the protection of the privacy of the research participants.

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
