# Peer review of "Assessment and Training of Perceptual-Motor Function: Performance of College Wrestlers Associated with History of Concussion"

_brainsci, 2024, doi:10.3390/brainsci14010068_

Round 1

Reviewer 1 Report

Comments and Suggestions for Authors

The article presents a compelling investigation into the use of immersive virtual reality (VR) for assessing and training perceptual-motor function in college wrestlers with a history of concussion. The study is well-structured, with clear objectives, a solid methodology, and comprehensive data analysis. The use of VR as a tool for both assessment and intervention is innovative and relevant to current technological advancements in sports science. However, there are some areas that require further consideration.

Strengths:

  1. Novelty and Relevance: The application of VR in assessing and training perceptual-motor skills in athletes with a history of concussion is a novel approach. It addresses an important gap in sports medicine and rehabilitation.
  2. Methodology: The experimental design, including pre- and post-training assessments, and the use of a control group, is robust. The inclusion of detailed procedural descriptions enhances the reproducibility of the study.
  3. Statistical Analysis: Comprehensive statistical methods, including ROC analyses, ANOVA, and sensitivity-specificity evaluations, add credibility to the findings.

Weaknesses:

  1. Sample Size and Diversity: The study focuses only on male wrestlers. Including female athletes or athletes from other sports might provide a more comprehensive understanding of the VR tool's applicability across different groups.
  2. Control Group Design: While a control group is used, the study could be strengthened by random assignment to control and experimental groups to minimize selection bias.
  3. Long-Term Effects: The study assesses short-term training outcomes. Long-term follow-up would be beneficial to understand the sustained impact of VR training on athletes.
  4. Technology Dependency: The reliance on specific VR technology, which may not be widely accessible, could limit the generalizability of the findings.

To enhance the quality and impact of the manuscript, the authors could consider the following improvements and inclusions:

  1. Detailed Participant Demographics: Include more detailed demographics of the participants (e.g., experience level, specific weight classes, previous concussion details). This information can provide a deeper context for interpreting the results and assessing their applicability.
  2. Methodological Clarifications:
    • Clarify the selection criteria for participants, particularly regarding their concussion history.
    • Detail the calibration and standardization process of the VR technology to ensure consistency across assessments.
  3. Technical Specifications and Limitations: Provide more details about the VR equipment and software used, including any limitations that might affect the results or their generalizability.
  4. Data on Training Intensity and Compliance: Include data on the intensity and compliance of the participants during the VR training sessions. This information can help in understanding the dose-response relationship.
  5. Broader Theoretical Framework: Integrate a more comprehensive theoretical framework that links the findings to broader concepts in neuroplasticity, concussion recovery, and athletic training.
  6. Discussion on Practical Implications: Expand the discussion section to address the practical implications of the findings for coaches, trainers, and sports medicine professionals.
  7. Addressing Potential Confounds: Discuss and control for potential confounding variables such as the athletes' training regimen outside the study, psychological factors, and other health-related variables.
  8. Limitations Section: Add a detailed limitations section that not only addresses the potential biases and constraints of the study but also suggests how these could be mitigated in future research.
  9. Ethical Considerations: Add a section discussing the ethical considerations of the study, especially given the involvement of human subjects with a history of medical conditions (concussions).
  10. References to Current Literature: Ensure the literature review is up-to-date and includes recent studies that have been published on related topics.
  11. Practical Recommendations for Implementation: Based on the study's findings, provide specific, actionable recommendations for implementing VR-based training in athletic programs, especially those with athletes who have a history of concussions.
  12. Supplementary Material: Consider adding supplementary material that could include detailed methodology, additional data analyses, or even video demonstrations of the VR training protocol.

Incorporating these improvements can enhance the manuscript's comprehensiveness, relevance, and utility for readers and practitioners in the field.

Comments on the Quality of English Language

Please proofread your manuscript. There are some minor grammatical/syntactic errors. 

Author Response

Weaknesses:

Sample Size and Diversity: The observational cohort study design necessarily limited participants to those who were members of the collegiate wrestling team. To further our understanding of possible sex, age, sport, and level of competition differences, we are currently analyzing the results of another observational cohort study that included 50 female high school soccer players. In the next to last paragraph of the Discussion section, we stated: “In addition to concussion history, important factors to consider that may influence training effects include sex, age, sport, and level of competition.”

Control Group Design: The study design assigned participants to groups based on self-reported history of concussion (confirmed by review of medical records), which produced a “comparison” group of participants who denied a history of concussion. The relatively small number of wrestling team members precluded further random subdivision into experimental (training) and control (no training) groups. Furthermore, coaches do not want any athletes assigned to a control condition that would deprive them of a potentially beneficial training effect. Acceptance of this constraint was necessary to conduct the study.

Long-Term Effects: We agree that long-term follow-up would be beneficial. In the next to last paragraph of the Discussion section, we stated: “The volume of perceptual-motor training required to realize maximum benefit remains unknown, as well as the frequency of training sessions required to retain performance gains.”

Technology Dependency: A sentence has been added to the last paragraph of the Discussion section: “Although an immersive VR system is necessary for clinical application of this study’s results, the cost of such equipment is far more affordable than the diagnostic neuroimaging methods that are currently required for detection of subtle impairment.”

Detailed Participant Demographics: The phrase “representing all weight classes” was inserted into the first sentence of the Participants subsection of the Methods section. The reported mean and standard deviation for age demonstrates that most of the participants ranged from 18 to 22 years of age. If the precise age range is needed, it can be inserted after the mean age and standard deviation. Alternatively, the median age and range can be reported. The first sentence of the Results section specified that the “median time since the most recent concussion occurrence was 22 months (range: 9 to 120 months).” The only additional concussion-related information collected was the number of prior concussions. The following content has been added: “a single concussion was reported by 6 and ≥ 2 concussions was reported by 4.”

Methodological Clarifications: As a cohort study, all members of the collegiate wrestling team were included as participants. A history of concussion was used to classify cases, but it was not a consideration for inclusion in the study. Content concerning calibration/standardization has been added to the beginning of the Procedures subsection: “Prior to testing, an eye calibration procedure ensured accurate measurement of eye position in relation to visual stimuli presented on the VR headset display (Pico Neo 3 Pro Eye, Pico Immersive, Ltd., 123 Mountain View, CA). A T-pose position (i.e., standing upright with both arms abducted to 90° and elbows fully extended) was used to acquire a measurement of the distance between hand controllers, which was used to calibrate the positions of left and right virtual response targets at 30% beyond maximum arm reach distance and outside the peripheral field of view (i.e., eye and neck movements required to visualize a virtual response target).”

Technical Specifications and Limitations: Manufacturer and model of the VR equipment has been specified.

Data on Training Intensity and Compliance: Content added to end of the last paragraph of the Procedures subsection: “Because the training was an element of scheduled team activities, 100% compliance was attained.”

Broader Theoretical Framework: The second paragraph of the Discussion section provides a quite broad framework for relating the study findings to brain network connectivity, and the third paragraph further addresses the relevance of behavioral performance consistency to variability in the neural signals conveying information within and between networks.

Discussion on Practical Implications: Content was added at the end of the third paragraph of the Discussion section to specify the practical implications more clearly: “Furthermore, our finding of significant post-training IIV decreases for both groups suggests that favorable neuroplasticity was induced by the immersive VR training program, which may promote resolution of long-term concussion effects, reduce risk of sport-related injury, and enhance sport performance capabilities.”

Addressing Potential Confounds: Because all the participants were members of the same collegiate wrestling team, physical activities performed outside the study’s training program can reasonably be assumed to have been highly consistent for each wrestler. Other potential confounding factors are mentioned in the content added to the paragraph addressing limitations.

Limitations Section: Content added to the last sentence of the paragraph on limitations: “Potential confounding factors that we did not address, such as sleep disruption, depression, anxiety, and any chronic health disorders, should be included as covariates in future research.”

Ethical Considerations: To provide elaboration, the following content was added to the Participants subsection paragraph: “The athletes were advised that participation was voluntary, rather than mandatory for all team members.”

References to Current Literature: Over 40% of the references (20/46) have been published since 2020 and 20% (9/46) have been published within the last 2 years.

Practical Recommendations for Implementation: To provide very specific, actionable recommendations, content has been added to the end of the Conclusions section: “Ideally, all athletes on a given team would be tested to identify any immersive VR performance deficiency that might be reduced through training that integrates perceptual, cognitive, and motor demands.”

Supplementary Material: Video of the events viewed on the VR headset display would be helpful to provide a thorough understanding of the immersive experience. Such a video is not currently available, but could be produced by the end of December or early January.

Reviewer 2 Report

Comments and Suggestions for Authors

While the paper addresses an important question and introduces a potentially valuable tool in VR training, some major points should be addressed before further consideration.

  1. The necessity of using immersive VR in this study should be more clearly justified. It is important to understand why VR is specifically required for the tasks performed in this study and whether similar outcomes could be achieved with simpler, non-VR tasks (e.g., button presses).
  2. The paper should present and compare the raw data for PL and RT, not just the IIV. This would provide a more comprehensive understanding of the results and their implications.
  3. The term “perceptual latency” used to describe the latency between stimulus onset and movement initiation is misleading, since this latency clearly include a motor element.

Minor points:

  1. Line 157: wrong format of the citation
  2. Line 196, HxSRC and NoSRC: inconsistent abbreivations

Author Response

Major points:

1. The Introduction section includes the following statements supported by cited references that provide justification for the use of immersive VR:

    • “…the best means to assess and train cognitive aspects of sport performance remains unclear [16].”
    • “Rather than simultaneously administering distinctly different cognitive and motor tasks, a visuospatial task that simultaneously engages perceptual, cognitive, and motor processes for achievement of a whole-body functional goal may have greater relevance to sport performance and injury resistance [12,13,15,16,22].”
    • “Immersive virtual reality offers a means to acquire numerous measurements during the performance of a perceptual-motor task that requires complex whole-body responses to visual stimuli [25,26].”
    • “An integrated multi-component VR assessment may identify individuals who would otherwise be exposed to an unrecognized elevation of injury risk [27], and participation in a training activity that incorporates a similar approach may yield beneficial improvements in perceptual-motor performance.”

  1. A Supplemental Table has been created to provide the requested data. Tables 2 and 3 include PL (Avg) and RT (Avg) data. Because PL (Avg) for arm, neck, and step movements each had a highly skewed distribution, the Geometric Mean (back-transformed value of the natural log of the original data, which closely estimates the median of the original data) is presented in Table 2. This procedure provides more accurate parametric analysis results for data that do not distribute normally. The corresponding RT (Avg) values demonstrated distribution normality, with mean and standard deviation values presented in Table 3.

  1. Yes, a motor element is included in the operational definition of perceptual latency. Some amount of movement is necessary to delineate its onset. The threshold displacement values (i.e., 6 degrees of neck rotation, 10 cm of hand controller displacement, or 10 cm of step displacement) were derived from testing that established the minimum displacements that consistently avoided “false” indications of a “full” movement response toward a left or right response target (i.e., elimination of the effect of small-amplitude back and forth oscillatory movements occurring prior to a “full movement response”).

Minor points:

  1. Error corrected.

  1. Error corrected.

Reviewer 3 Report

Comments and Suggestions for Authors

I would like to congratulate the authors of this study, as what they present in this paper is very interested.

The introduction is complete and clear, well thought out and helps to understand the topic of study. The objective is well written and precise.

With regard to the methodological section, it is clear and very well explained. Although the sample is small, and even more so when it is divided between the group that has suffered a concussion and the group that has not.

I think that in the section on participants you should add information on the participants who have suffered a shock and the time that has elapsed since the shock. This is a factor that has been taken into account in the results and should therefore be explained more clearly in the method.

In the results section, I believe that because the sample size is small, the calculation of the effect size should be included, as this would help to see whether the statistically significant results have a greater or lesser effect on the variables evaluated.

The discussion section is correct, well understood and includes topical research.

The references used are relevant and up to date.

Author Response

The following content was added: “Among the 42% of wrestlers who self-reported HxC (10/24), a single concussion was reported by 6 and ≥ 2 concussions was reported by 4.” The amount of time elapsed since the most recent concussion occurrence was reported in the second sentence of the Results section: “The median time since the most recent concussion occurrence was 22 months (range: 9 to 120 months).”

Both Receiver Operating Characteristic Area Under Curve (AUC) and the Odds Ratio (OR) reported in Table 1 are considered effect sizes. The partial eta-squared values reported for Group X Session Interaction, Group Difference, and Session Difference in Table 2 and Table 3 also represent effect sizes associated with parametric repeated measures ANOVA results.

Round 2

Reviewer 1 Report

Comments and Suggestions for Authors

I thank the authors for taking time and effort to address my comments. I believe that the manuscript may be now published. 

Reviewer 2 Report

Comments and Suggestions for Authors

The authors have addressed all my concerns.